# Advances in Gluten Hypersensitivity: Novel Dietary-Based Therapeutics in Research and Development

**DOI:** 10.3390/ijms25084399

**Published:** 2024-04-16

**Authors:** Rick Jorgensen, Shambhavi Shivaramaiah Devarahalli, Yash Shah, Haoran Gao, Tamil Selvan Arul Arasan, Perry K. W. Ng, Venugopal Gangur

**Affiliations:** 1Food Allergy and Immunology Laboratory, Department of Food Science and Human Nutrition, Michigan State University, East Lansing, MI 48824, USA; jorgen70@msu.edu (R.J.); devaraha@msu.edu (S.S.D.); shahyash0707@gmail.com (Y.S.); gaohaora@msu.edu (H.G.); arultami@msu.edu (T.S.A.A.); 2Cereal Science Laboratory, Department of Food Science and Human Nutrition, Michigan State University, East Lansing, MI 48823, USA; ngp@msu.edu

**Keywords:** gluten hypersensitivity, wheat, allergy, IgE, enzyme hydrolysis, therapeutics, fermentation, preclinical test, clinical test, animal models, translational research

## Abstract

Gluten hypersensitivity is characterized by the production of IgE antibodies against specific wheat proteins (allergens) and a myriad of clinical allergic symptoms including life-threatening anaphylaxis. Currently, the only recommended treatment for gluten hypersensitivity is the complete avoidance of gluten. There have been extensive efforts to develop dietary-based novel therapeutics for combating this disorder. There were four objectives for this study: (i) to compile the current understanding of the mechanism of gluten hypersensitivity; (ii) to critically evaluate the outcome from preclinical testing of novel therapeutics in animal models; (iii) to determine the potential of novel dietary-based therapeutic approaches under development in humans; and (iv) to synthesize the outcomes from these studies and identify the gaps in research to inform future translational research. We used Google Scholar and PubMed databases with appropriate keywords to retrieve published papers. All material was thoroughly checked to obtain the relevant data to address the objectives. Our findings collectively demonstrate that there are at least five promising dietary-based therapeutic approaches for mitigating gluten hypersensitivity in development. Of these, two have advanced to a limited human clinical trial, and the others are at the preclinical testing level. Further translational research is expected to offer novel dietary-based therapeutic options for patients with gluten hypersensitivity in the future.

## 1. Introduction

The major source of dietary gluten in the human diet is wheat as it is one of the three main staple crops besides rice and corn, consumed globally, with the expected consumption to increase by 11% by 2031 [1]. Besides serving as a nutritional source, gluten is among the major food allergens regulated by multiple countries including the USA, Canada, the European Union, the United Kingdom, Australia, Japan, and New Zealand. However, because of its distinctive viscoelastic properties, gluten plays a vital role as a thickener and structure holder in food matrices, thus making it very challenging to exclude gluten from individuals’ daily diets [2,3,4].

Gluten allergens are seed storage proteins that are comprised of gliadins and glutenins. Aside from these gluten allergens, wheat also contains non-gluten allergens known as albumins (water-soluble) and globulins (saline-soluble), which have metabolic and structural functionalities in wheat plant biology [5]. Gliadins are prolamin proteins that are ethanol-soluble, and glutenins are glutelin proteins that are soluble in weak acid (acetic acid) solution [6]. Both gliadin and glutenin allergens are linked to gluten hypersensitivity (or allergy) in humans [7,8]. Non-gluten allergens can also elicit similar types of disease.

Based on the electrophoretic mobility and the similarity of amino acid sequences, gliadin allergens can be classified into three major subtypes: α-gliadins, γ-gliadins, and ω-gliadins [9,10,11]. All three types of gliadin allergens can elicit hypersensitivity reactions. Unlike the monomeric units of gliadin, glutenin allergens are proteins with multiple linked components held together via disulfide bonds. These glutenin subunit polymers possess viscoelastic properties. They are insoluble in water–alcohol mixtures unless these bonds are broken under specific conditions [11]. Based on their electrophoretic mobility on sodium dodecyl sulfate-polyacrylamide gel electrophoresis (SDS-PAGE), glutenin allergens can be further divided into high-molecular-weight glutenin subunits (HMW-GSs) (70 to 120 kDa) and low-molecular-weight glutenin subunits (LMW-GSs) (30–45 kDa) [12,13,14,15,16]. Despite making up approximately 10% of gluten protein, the HMW-GSs play a significant role in the end use quality [10,16]. Both ω-5 gliadin and HMW-GSs have been shown to be major allergens associated with wheat-dependent exercise-induced anaphylaxis (WDEIA) [17]. Many gluten-allergic individuals have been shown to be sensitized to both HMW-GSs and LMW-GSs [7,18,19]. Thus, both gliadins and glutenins are major gluten allergens that are naturally present in wheat that can elicit hypersensitivity reactions in humans.

Gluten proteins can elicit multiple immune-system-mediated diseases in humans. These include not only gluten hypersensitivity (commonly called allergy) but also celiac disease and non-celiac gluten sensitivity (NCGS) [6]. Gluten hypersensitivity is characterized by the production of IgE (Immunoglobulin E) antibodies against specific gluten proteins (i.e., allergens) and a myriad of clinical allergic symptoms including life-threatening systemic anaphylaxis (Figure 1) [20]. Celiac disease is an enteropathy with autoimmune characteristics and is triggered by gluten-containing foods in susceptible individuals that possess the human leukocyte antigen (HLA)-DQ2 and/or HLA-DQ8 haplotypes [21,22]. NCGS is described as a condition in which individuals experience distress after consumption of gluten but does not show characteristics of the other two conditions [6]. Improvements are experienced after following a gluten-free diet. Notably, only gluten hypersensitivity, but not the other two diseases, is mediated by gluten-specific IgE antibodies, and it is the focus of the research in this study [6,23].

When genetically susceptible individuals produce IgE antibodies against one or more gluten proteins, they are deemed to be sensitized to gluten (Figure 1). However, gluten-sensitized subjects typically do not have clinical symptoms of disease unless they are subsequently exposed to gluten via the oral, nasal, ophthalmic, or dermatologic routes (Figure 1). Currently, antihistamines, steroids, and epinephrine (adrenaline) are the only treatments recommended to manage clinical symptoms of allergic reactions to gluten, as is done for other food allergies such as peanut allergy. Notably, for life-threatening systemic anaphylaxis, epinephrine is the only life-saving emergency medication available at present [20].

Currently, the only recommended dietary-based treatment for gluten hypersensitivity is the complete avoidance of exposure to gluten. However, avoiding gluten exposure is extremely challenging given the widespread use of wheat in foods, feed, and cosmetics including skin care products. Nevertheless, extensive efforts are underway towards creating novel dietary-based therapeutics for gluten hypersensitivity disorders [24]. Here, we provide a comprehensive and up-to-date review of preclinical and clinical studies testing the novel dietary-based therapeutic approaches in animal models, and gluten-hypersensitive human subjects.

There were four objectives for this study: (i) to compile the current understanding of the mechanism of gluten hypersensitivity, including the roles of genetic and environmental factors; (ii) to critically evaluate preclinical testing of novel dietary-based therapeutics in animal models; (iii) to determine the potential of such novel therapeutics currently under research and development in gluten hypersensitive humans; and (iv) to synthesize the outcomes from these studies and identify gaps in the research to inform future translational research. We employed various combinations of the following keywords for our search using the PubMed and Google Scholar databases: gluten, hypersensitivity, therapy, in vivo, in vitro, IgE, wheat, hypoallergenicity, animal model, dog, rat, guinea pig, mice, human, and dietary-based. The PubMed search retrieved articles ranging from 2 to 1209; the Google Scholar search retrieved hits ranging from 919 to 1920. Only relevant English language articles were retrieved and used to address the above objectives. All articles chosen for the study are included in the references. The focus of this study was specifically on IgE-mediated gluten hypersensitivity, necessitating the exclusion of articles addressing non-IgE-mediated gluten disorders (including celiac disease, non-celiac gluten sensitivity, and eosinophilic disorders). Our findings collectively demonstrate that there are at least five promising dietary-based therapeutic approaches for mitigating gluten hypersensitivity in research and development. Of these, two approaches have advanced to a limited human clinical trial, and the others are at the preclinical testing level. Further translational research is expected to offer dietary-based therapeutic options for patients with gluten hypersensitivity.

## 2. Mechanism of Gluten Hypersensitivity

Gluten hypersensitivity is clinically classified as an immediate (or Type-I) hypersensitivity reaction mediated by the immune system in response to specific gluten proteins known as gluten allergens. Mechanisms of gluten hypersensitivity are incompletely understood at present. However, akin to other types of food allergies, it is thought to develop in two sequential phases (Figure 1): (i) the first phase of sensitization, where genetically susceptible individuals produce IgE antibodies specific to gluten upon initial exposures only under environmental conditions that are incompletely understood; once produced, these IgE antibodies attach to mast cells in the tissues and to basophils in the blood via the high-affinity IgE receptor; such gluten-sensitized subjects do not have clinical symptoms of disease when they are not exposed to gluten (ii) The second phase, known as clinical elicitation of allergic reactions, occurs when sensitized individuals exhibit clinical symptoms of allergic reaction upon oral or other routes of re-exposure to gluten. Among the gluten hypersensitivity reactions, systemic anaphylaxis is potentially fatal and requires emergency medical management to save life. In case of WDEIA, exercising within 1–4 h upon consuming gluten results in systemic anaphylaxis [25,26].

Specific genetic and environmental factors leading to the genesis of gluten hypersensitivity are incompletely understood at present. However, emerging evidence shows that it is a complex genetic disorder like other food allergies, and it involves both susceptible immune gene variants as well as immune-modulating environmental factors, both of which are beginning to be unraveled. Recent research has identified several genetic factors as potential risk factors for developing gluten hypersensitivity disorders (Figure 1). These include the genes encoding for the skin immune barrier function gene (filaggrin), genes involved in allergen presentation by antigen-presenting cells such as dendritic cells, macrophages/monocytes and B cells (MHC class II genes), the T helper (Th)-2 immune response regulator genes (IL-4, and IL-4 receptor) that are required for developing IgE antibody production, and genes encoding proinflammatory cytokines (Il-18) and innate immune receptors (TLR4) [27,28,29,30,31,32,33,34,35]. Such genetic variants work in concert with poorly known environmental factors to create conditions for the development of gluten hypersensitivity disorders. Currently, the following four types of environmental factors are implicated: (i) early-life exposure to pets such as cats—interestingly, such exposure appears to offer protection from developing gluten hypersensitivity; (ii) deficiency of vitamin D, a major immune-function-modulating nutrient, has been shown to increase the risk for developing gluten hypersensitivity; (iii) gut microbial composition has a major impact on food allergy development including gluten hypersensitivity; and (iv) the use of antacids/antiulcer mediations has been shown to increase the risk of developing gluten hypersensitivity (Figure 1) [36,37,38,39,40].

During the first phase, susceptible individuals are exposed to gluten via various routes: eyes, nose, skin, and mouth. These allergens are captured by antigen-presenting cells (i.e., macrophages and dendritic cells), processed and presented to T cells. Coupled with several co-factors such as a dysregulated host microbiome and other environmental factors (e.g., detergents in allergen-containing cosmetic products), primed Th-2 cells in sensitized subjects activate B cells to produce allergen-specific Immunoglobulin E (IgE) antibodies. These allergen-specific IgE antibodies then bind to the high-affinity IgE receptor (FcεRI) present on the surface of mast cells and basophils. Upon re-exposure of sensitized individuals, gluten allergens cross-link the IgE on mast cells and basophils and activate them to release histamine and other mediators [20]. These mediators cause clinical symptoms of allergic disease.

Depending on the dose and route of exposure, symptoms of allergic reactions vary from rashes, hives, vomiting, diarrhea, airway hyper-responsiveness, and conjunctivitis to severe reactions such as life-threatening systemic anaphylaxis and asthma attacks known as baker’s asthma. Individuals who suffer from allergic rhinitis and conjunctivitis exhibit increased mucus secretion, itching and sneezing [41]. Gluten allergens, when they enter the bloodstream upon ingestion, can cause systemic allergic reactions known as systemic anaphylaxis, involving multiple organs including the gut, skin, heart, and lungs. Such reactions are capable of causing airway constriction such as difficulty in breathing (asthma attacks) and severe hypotension resulting in anaphylactic shock, which can be deadly [42,43]. Exercising within 1–4 h after gluten consumption can cause WDEIA in sensitized subjects. Late-phase reactions can occur 6–8 h after the initial immediate reaction due to new mediators’ release by mast cells/basophils and may persist for up to 24 h [44].

## 3. Preclinical Development of Novel Dietary-Based Therapeutics Using Animal Models of Gluten Hypersensitivity

Several novel therapeutic approaches are being developed using canine and rodent models of gluten hypersensitivity for future clinical application in humans. These are reviewed below.

### 3.1. Animal Testing 

Animal models for gluten hypersensitivity were first developed using a dog model, and then subsequently using rat, guinea pig, and mouse models. Here, several animal models are systematically evaluated for potential development of immunotherapies to combat gluten hypersensitivity. 

#### 3.1.1. Canine Model: Potential of Thioredoxin and Heat-Killed *Listeria monocytogenes* to Reduce Gluten Allergenicity

Using inbred dogs, a novel animal model of food allergy including gluten hypersensitivity was developed and used to test the potential of novel therapeutics [45,46]. In 1997, Buchanan et al. aimed to investigate whether altering the biochemical and physical properties of wheat proteins including gluten through reduction with thioredoxin could impact their allergenic properties [45]. They used inbred high-IgE-responder dogs (spaniel/basenji) and developed a complex protocol to study the impact of thioredoxin treatment on the molecular nature of gluten allergenicity (Table 1). They found that thioredoxin treatment increased the amount of gluten protein required to induce skin allergic reaction upon skin injection, suggesting the positive outcome of reduced gluten allergenicity. A potential limitation of this study would be the administration of the allergen via subcutaneous sensitization but not the oral route to elicit allergic reaction. Further research is needed to address this limitation and validate the potential of thioredoxin treatment to reduce or eliminate oral gluten allergenicity.

Another study using the same canine model explored the use of heat-killed *Listeria monocytogenes* (HKL) as a potential therapy [46]. Briefly, subcutaneous immunization of dogs was conducted with a mixture of wheat and cow’s milk extract in conjunction with HKL (Table 1). The findings revealed that as the immunizations progressed, the mean dose required to trigger a positive skin prick test increased, suggesting the potential effectiveness of this therapy in mitigating gluten allergic responses. However, they did not study the effect of HKL on oral gluten-induced allergic reactions. Furthermore, whether HKL would be an acceptable dietary component for administering with gluten in humans remains to be investigated. This warrants further investigation and refinement in future studies focusing on these two general issues.

#### 3.1.2. Evidence from Rat Models: Potential of Gluten Genetic Deletion, Deamidation, and Enzymatic Digestion to Reduce Gluten Allergenicity

In rat models, there have been several studies exploring potential therapeutics for gluten hypersensitivity (Table 2). One such study aimed to investigate the allergenicity of 1BS-18 (deletion in the 1B short arm) Hokushin wheat (a wheat variety deficient in ω-5 gliadin gene) utilizing a rat model of anaphylaxis as quantified by hypothermic shock responses (HSR) [47]. They found that rats sensitized with ω-5 gliadin experienced significantly lower HSR when challenged with 1BS-18 gluten that is deficient in ω-5 gliadin as compared to control Hokushin gluten containing ω-5 gliadin. They also found that 1BS-18 gluten-sensitized rats exhibited no HSR when challenged with ω-5 gliadin, whereas those sensitized with Hokushin gluten and challenged with ω-5 gliadin displayed significant HSR. There are some limitations to this model: (i) the reported significant changes in HSR were not substantial; (ii) the use of adjuvant limits generalization on intrinsic allergenicity interpretations on the gene deficient wheat line; and (iii) the very low reported specific IgE optical densities (OD) in sensitized mice (stated as OD of 0.09) may raise questions about the study’s robustness and applicability to real-world scenarios.

A subsequent study aimed to determine whether gluten prepared from 1BS-18 wheat would induce oral tolerance (OT) to both gluten and ω-5 gliadin [48]. The findings revealed that gluten-specific IgE decreased in OT-induced rats that received 1BS-18 gluten. Moreover, hypersensitivity reactions were not observed in OT-induced rats challenged with 1BS-18 gluten. Similarly, ω-5 gliadin-specific IgE levels decreased in OT-induced rats that received 1BS-18 gluten, and hypersensitivity reactions were not observed when challenged with ω-5 gliadin. The study’s strength includes its exploration of the potential of 1BS-18 gluten to establish OT to both gluten and ω-5 gliadin, which has implications for understanding and managing wheat allergies. However, the research did report minimal changes in rectal temperature and very low optical density for specific IgE antibody levels, potentially raising questions about the robustness of the results. Combining these findings with the previously reported study by Yamada et al. (2019) [47] indicated the lower allergenicity of 1BS-18 gluten, offering promising insights into the development of a ω-5 gliadin-deficient line for allergy management.

The potential of deamidation as a method to reduce gluten allergenicity has been tested in rats [49]. The study found that deamidated gliadin had the potential to induce oral tolerance in rats. The positive impact of IgE responses was reported. However, the low optical density values for specific IgE antibody seen in rat testing raises questions regarding the robustness of their findings. They did not study disease phenotypes of gluten hypersensitivity.

Another study in rats aimed to investigate whether hypoallergenic gluten containing wheat flour could be created to mitigate the effects of airway inflammation associated with gluten allergy [50]. The hypoallergenic flour was prepared by digesting the wheat flour with cellulase and actinase. The research revealed that hypoallergenic gluten elicited lower IgE levels. Reduced immune cell counts (eosinophils, lymphocytes, and neutrophils) were observed in the broncho-alveolar lavage fluid after intranasal challenge with the modified gluten. The merits of this research demonstrate the potential of hypoallergenic wheat to induce oral tolerance, subsequently reducing allergy-related immune cell counts in the airways. However, it would be interesting to assess the intrinsic allergenicity of this novel hypoallergenic gluten using adjuvant-free mouse models of gluten hypersensitivity and in food-allergy-related disease phenotypes.

#### 3.1.3. Guinea Pig Model: Potential of Gluten Genetic Deletion to Reduce Gluten Allergenicity

In guinea pigs, Kohno et al. utilized the 1BS-18 wheat line lacking the ω-5 gliadin locus to determine if it was capable of causing sensitization [51] (Table 2). The findings revealed that the allergic scores associated with WDEIA (commonly associated with ω-5 gliadin) were significantly lower in the 1BS-18 line. However, while the decreased allergic scoring is promising, the notable lack of specific IgE data, which is crucial when assessing the sensitization capacity of wheat, is concerning. Additionally, the study could benefit from providing more detailed information on the scoring system used in the challenge test. Further research is needed to test the effect of this gene targeted wheat on other food allergenicity-related phenotypes.

#### 3.1.4. Mouse Models: Potential of Probiotics, Enzymatic Digestion, L-arabinose, Deamidation, Polyphenols from Fruits, and Phosphorylation of Gluten to Reduce Wheat Allergenicity

Several mouse model studies explored six novel dietary-based therapies in gluten hypersensitivity (Table 3). One such study by Fu et al. explored the effects of *Pediococcus acidilactici* XZ31 in mitigating gluten allergenicity [52]. Fermentation with *P. acidilactici* promotes the digestion of gluten by pepsin and trypsin, reducing antigenic reactions. The findings indicated that *P. acidilactici*-treated gluten showed decreased IgE as well as anaphylactic scores when compared to the control gluten sensitized and challenged group. Histological scores within the duodenum showed no differences between treated vs. control groups. While the study demonstrated several strengths, such as its ability to highlight the regulation of Th1/Th2 imbalance and lower gluten specific IgE levels, it also revealed some weaknesses, notably the lack of effects on histology scores, which warrants further examination.

Another study in mice aimed to assess the allergenicity of gluten derived from hydrolyzed wheat flour (HWF) treated with *Aspergillus niger*-fungi-derived prolyl endopeptidase (AN-PEP) compared to the standard wheat gluten [53]. The study found positive effects of reduced IgE by AN-PEP-HWF. Furthermore, various inflammatory markers including INF-γ, TNF-α, IL-4, IL-6, and IL-15 were elevated in mice exposed to regular wheat flour, but not in those exposed to AN-PEP-HWF. This paper demonstrates the potential of AN-PEP as a novel treatment method to produce hypoallergenic gluten. However, the one weakness is the absence of gluten hypersensitivity disease phenotype, which should be addressed in future work.

Another study sought to investigate the antiallergic activities and underlying mechanisms of L-arabinose in a mouse model of gliadin hypersensitivity [54]. They found that the anaphylactic scores in the group receiving L-arabinose and gliadin were lower than those mice receiving gliadin. The IgE levels and histamine levels were also lower, suggesting a positive outcome of treatment. Therefore, L-arabinose emerges as a candidate therapy. However, a notable weakness would be the use of alum adjuvant, which does not allow for the testing of the intrinsic allergenicity potential of gluten. Additionally, while the histamine levels did decrease, the reported levels in this model were notably low when compared to other models, suggesting a milder type of gluten hypersensitivity reaction observed in this model [57].

Abe and co-workers investigated the effects of deamidation of gliadin on allergenicity in a mouse model [57]. Oral administration of gliadin or deamidated gliadin was compared between mice sensitized with gliadin in the presence of alum adjuvant. The results elucidate that deamidation lowered histamine levels, decreased IgE levels, and reduced intestinal permeability. While this study provides another promising approach via the utilization of deamidation of gliadins, it is important to acknowledge its limitations. The use of alum adjuvant does not permit the evaluation of effects of treatment on the intrinsic gluten allergenicity. Additionally, the study reported very low optical densities for IgE levels, which may raise concerns regarding the robustness of the results.

Another study investigated the interaction between plant polyphenol extracts with gliadins to reduce gluten allergenicity [58]. The study used Balb/cJ mice sensitized with gliadin. In this model, cranberry was the only polyphenol to decrease gliadin recognition by both IgG and IgE antibodies as well as prevent the degranulation process in mast cells. However, the use of alum adjuvant limited the exploration of the effects on the intrinsic allergenicity of gluten. Disease phenotypes were not studied.

Xue and co-workers sought to treat gliadins with phosphorylation, alcalase, and papain hydrolyses to determine the effects on gliadin allergenicity [56]. Mice were sensitized with either native gliadin, phosphorylated gliadin, hydrolyzed gliadin with alcalase, or hydrolyzed gliadin with papain via intraperitoneal injections. Total IgE, specific IgE, histamine, and select cytokines were measured. Mice sensitized with treated gliadin exhibited significantly lower levels of total and specific IgE. Histamine, serum IFN-γ, and serum IL-4 were decreased in the mice exposed to treated gliadin when compared to non-treated gliadin. Thus, phosphorylation and alcalase or papain digestion appear as promising ways to reduce gluten allergenicity.

Li and co-workers investigated the use of pepsin- and trypsin-treated gluten to reduce the adverse effects suffered during oral immunotherapy [55]. In one portion of the study, the IgE binding capacity in mice sensitized with gliadin via intraperitoneal injections was shown to have a lower affinity for any of the treated gliadins. In the oral tolerance model, mice received oral administration of gliadin- or pepsin-treated gliadin followed by subsequent intraperitoneal immunization. The oral challenge was conducted with gliadin. The gliadin-specific IgE and IgG1 were significantly decreased by oral administration of gliadin- or pepsin-treated gliadin, suggesting the development of oral immune tolerance. Disease phenotypes were not studied.

Thus, in several mouse models, novel dietary-based therapeutic approaches are being researched and developed for potential future human application. However, in future research more disease phenotypes need to be considered, and the use of adjuvant-free models are needed to interpret the impacts of treatment on intrinsic gluten hypersensitivity reactions.

## 4. Other Approaches to Produce Hypoallergenic Gluten Products

Tsurunaga et al., 2023, recently released an exciting report [59]. They sought to investigate the potential reduction in wheat allergenicity through the addition of tannins from chestnut inner skin (CIS) or young persimmon fruit (YPF) to wheat flour. Cookies made with cake flour were prepared, with tannins accounting for 3%, 5%, and 10% of the total ingredient weight. Both CIS and YPF were chosen for their high tannin content. The evaluation of wheat allergen content involved the use of two ELISA kits, demonstrating a notable reduction in protein content, especially gliadin and other wheat proteins, with CIS treatment showing more marked effects than YPF treatment. Immunoblotting with a polyclonal rabbit anti-ω5-gliadin IgG antibody further supported these findings, revealing decrease in IgG immunoreactivity in both types of treatments. Despite the strength of demonstrating reduced immunoreactivity, this study has certain limitations. IgE binding was not investigated, and considering the IgE-mediated nature of wheat allergy, further research is essential in this regard to determine the effects on allergenicity. Additionally, the immune reactivity study focused solely on ω-5 gliadin, necessitating future exploration of the tannin treatment effects on other wheat allergens to comprehensively understand its potential impact. Furthermore, future in vivo testing using preclinical animal models will be necessary as a crucial step preceding eventual human clinical trials [60,61]. 

Yu et al., 2023, investigated whether it was feasible to modify specific segments of gluten-encoding genes to substantially diminish wheat immunotoxicity without adversely affecting the physiochemical properties crucial for breadmaking [62]. The hypothesis, though implicit, postulated that multiplexed CRISPR-Cas9 editing of the common wheat (*Triticum aestivum*) Fielder cultivar (originally released by the University of Idaho in 1974), aimed at modifying ω- and γ-gliadin genes, would result in reduced immunoreactivity compared to the non-edited counterpart. To implement this approach, seven gRNAs were designed to edit the gliadin gene, with three targeting ω-gliadins on chromosomes 1A and 1D and four targeting ω-gliadins on chromosome 1B. Monoclonal antibodies R5 and G12, known for their high predictability of wheat immunotoxicity in gluten-sensitive patients, were employed to assess immunotoxicity through ELISAs. The findings indicated a significant reduction in immunoreactivity in the edited wheat line compared to the non-edited line. Despite the strength of demonstrating reduced immunoreactivity, the need for further investigations remains, particularly examining IgE reactivity (allergenicity) in these edited lines. Further in vivo testing in preclinical animal models would allow for the evaluation of CRISPR-Cas9 editing’s effect on potentially reduced clinical allergic reactivity. This step is essential before progressing to clinical trials involving human subjects [60,61].

## 5. Effects of Fermentation, Gluten Gene Targeting, Deamidation, Thioredoxin, and Enzyme Treatment on Gluten Allergenicity in Humans

Several therapeutic approaches discussed in the previous section have been applied to human gluten hypersensitivity. Most are at the preclinical testing stage, with two studies at a limited clinical testing stage (Table 4).

### 5.1. Soy Sauce Fermentation Can Reduce/Eliminate Human Gluten Allergenicity: In Vitro Evidence

Traditionally, soy sauce is produced using wheat and soybeans used in a 1:1 ratio. There is extensive evidence that the fermentation process in general has the potential to reduce or eliminate the allergenicity of food proteins [63]. In particular, seminal studies by Japanese investigators demonstrate the allergenicity reducing/eliminating power of soy sauce fermentation processing [63]. During soy sauce fermentation, molds, yeast, and bacteria work on soy and wheat proteins to create the final product. Depending on the method of soy sauce fermentation used in different countries, different types of molds, yeasts, and bacteria have been identified in the process of soy sauce production (Table 5). Previous studies examined the effect of soy sauce fermentation after every major step in the production process. For example, the effects on allergenicity after roasting, mold treatment, yeast and bacterial treatments have been demonstrated (Figure 2). Major findings from these studies are illustrated below.

Kobayashi and co-workers studied the effects of soy sauce fermentation on wheat protein including allergenicity in vitro using ELISA [63]. They reported a progressive decrease in detectable salt-soluble (non-gluten) as well as salt-insoluble (gluten) allergens during soy sauce fermentation processing (Figure 2). They used pooled serum from five wheat-allergic children as a source of anti-wheat IgE antibodies in their assay. 

Salt-soluble non-gluten allergens were present during the first two stages (raw material and *koji*) of production at comparable levels, suggesting no marked effect of mold treatment during the *koji* stage. However, a dramatic reduction in salt-soluble wheat allergens was noted progressively during the *moromi* stage of fermentation (with yeast and lactic acid bacteria) with 50% of the allergens lost by day 48 and over 91% lost by day 67 of *moromi* fermentation. Interestingly, no allergens were detectable in the raw soy sauce to the final product.

They also investigated the effect of soy sauce fermentation on salt-insoluble (gluten) allergens using a direct IgE ELISA method. They found that, as opposed to salt-soluble allergens, gluten allergens were decreased during the *koji* stage of production because of their solubilization due to enzymatic degradation. Furthermore, by day 10 of the *moromi* stage, nearly all the gluten proteins were undetectable, suggesting that gluten proteins are more susceptible to degradation by mold (*koji* stage), yeast and bacteria (*moromi* stage) than the salt-soluble wheat allergens. Nevertheless, both gluten and non-gluten allergens were absent in raw soy sauce, suggesting that the final product may be hypo/non-allergenic with complete degradation of both types of wheat allergens.

Researchers have extensively examined the microorganisms involved in soy sauce fermentation (Table 5). In general, soy sauce fermentation involves the first step of mold activity leading to the formation of *koji*. This is followed by yeast and lactic acid bacterial fermentation during the *moromi* stage. Depending on the country of origin, different groups of molds, yeasts, and bacteria have been identified in soy sauce fermentation process (Table 5). However, this research shows that there are some common genera of molds, yeasts, and bacteria independent of the country of origin producing the soy sauce (Table 5, see bold-faced microbes). 

These studies together demonstrate that (i) microbial fermentation of wheat has the potential to reduce and possibly even eliminate the allergenicity of both non-gluten and gluten proteins; (ii) elucidating the specific effects of mold versus yeast versus bacteria on wheat gluten versus non-gluten protein allergenicity urgently needs further investigation; and (iii) since the microbial composition of soy sauce produced in different countries can significantly differ, precise identification of microbes responsible for reducing and/or eliminating wheat gluten and non-gluten allergenicity is required.

### 5.2. Gluten Gene Targeting Can Be Used to Develop Hypoallergenic Wheat Lines: In Vitro Evidence from Human Studies

Several in vitro models have investigated the effects of the deletion of specific genes to engineer potential hypoallergenic wheat lines. One such study by Denery-Papini et al. explored the genetic variability at the Gli-B1 locus (responsible for encoding ω-5 gliadins), and how this would affect the responsiveness of IgE antibodies in individuals with WDEIA and urticaria [64]. One of the wheat cultivars, Clément (produced via replacing the short arm of the 1B chromosome with the portion of the short arm of the 1R chromosome from rye; 1BL/1RS), yielded minimal/no IgE reactivity in immunoblotting with serum from WDEIA and Urticaria patients. However, it did react with reduced binding with serum from an individual suffering from anaphylaxis. Thus, the 1BL/1RS translocation may reduce wheat allergenicity.

Waga et al. (2014) investigated the IgE binding capacity of glutens extracted from a wheat line with inactivated gene variants in the three gliadin containing loci (Gli A1, Gli B1, and Gli D1) by traditional plant breeding. Sera from wheat allergic patients (*n* = 10, specific type of allergy not specified) was used in ELISA testing as the source of IgE antibodies [65]. A significant decrease (approximately 30%) in IgE binding compared to the control wheat was noted. Further in vivo testing is needed to evaluate the allergenicity of this new wheat line. 

Lee et al. (2022) elucidated the use of a deletion line (ω5D) as a method for reducing allergenicity in gluten [67]. This new cultivar has selective deletions in the 1B chromosome Glu-B3 locus, thus causing it to lack ω-5 gliadin as well as some LMW glutenins and γ-gliadins [67]. They used serum from 14 WDEIA patients and 7 classical wheat allergy patients for testing IgE reactivity using ELISA and inhibition CAP system. They found a significant reduction in IgE binding of gliadins and glutenins from this line as measured by immunoblotting and inhibition ELISA. They concluded that oral challenge testing is needed to confirm the potential hypo-allergenicity of this line in WDEIA patients. They did not discuss the relevance of their findings regarding classical wheat allergy patients.

Altenbach and co-authors sought to explore the use of a transgenic wheat line utilizing RNA interference to silence the ω-5 gliadin gene as a potential hypoallergenic wheat line [66]. Sera from patients suffering from WDEIA (and some with urticaria or rhinitis; *n* = 11) was used in IgE immunoblotting. A reduction in IgE binding was seen in the transgenic lines when compared to a traditional wheat line. However, while a reduction in IgE reactivity was noted in the transgenic wheat lines, it was not eliminated completely; thus, further in vivo testing is needed to evaluate the allergenicity of this wheat line.

Waga and co-workers investigated the effects of thioredoxin on both immunoreactivity and dough rheological properties in ten winter wheat genotypes [68]. Sera from patients suffering from baker’s asthma (*n* = 2), chronic atopic dermatitis and food intolerance (*n* = 2), and chronic urticaria and angioedema (*n* = 1) was used in a direct and sandwich ELISA. Both ELISAs revealed a reduction (>50%) in IgE binding in all wheats treated with thioredoxin when compared to the native samples. Treatment with thioredoxin did not significantly impact dough rheological properties, suggesting thioredoxin as a promising method for reducing allergenicity without impacting wheat quality.

### 5.3. Gluten Gene Targeting and Enzyme Hydrolysis to Develop Hypoallergenic Wheat Products: Clinical Evidence from Testing in Gluten Hypersensitive Subjects

A naturally ω-5-gene-encoding chromosome-B-deficient ancient wheat known as *Triticum monococcum* (Einkorn, AA genotype) offers a unique opportunity to develop hypoallergenic wheat products for ω-5 gliadin allergic subjects such as WDEIA patients where ω-5 gliadin is the major allergen. Lombardo et al. sought to study the immunoreactivity of proteins from *Triticum monococcum* (Einkorn, AA genotype), a diploid ancestral wheat lacking the B chromosome; notably, ω-5 gliadin (the major wheat allergen in WDEIA) is encoded on the B chromosome [69]. Using skin prick testing with both *Triticum monococcum* and *Triticum aestivum* (commercial wheat, AABBDD genotype) revealed no positive reactions in *Triticum monococcum*, whereas 43% of WDEIA patients tested positive when exposed to *Triticum aestivum*. Patient sera revealed a lack of IgE immunoreactivity to ω-5 gliadin observed in *Triticum monococcum*. However, future studies testing oral reactivity are needed to confirm the suitability of Einkorn wheat for WDEIA patients.

Watanbe and co-workers have spent a great deal of effort to successfully develop a novel dietary-based treatment for gluten hypersensitivity. They first created a hypoallergenic wheat flour using both cellulase and actinase as hydrolyzing agents [77]. Later, they investigated the safety of the hypoallergenic wheat flour by creating a hypoallergenic cupcake and testing it in children with gluten-induced atopic dermatitis (AD) [70]. They reported that, upon consumption, 13 of the 15 children showed no adverse response, with only two of the patients showing an immediate reaction in the form of severe urticaria [70]. Interestingly, more than half of the patients were able to consume normal wheat products after consuming the cupcakes over a period of more than six months, suggesting the development of oral tolerance. Therefore, the consumption of such hypoallergenic wheat products offers a promising method for creating oral immunotolerance in individuals suffering from gluten-induced AD type of hypersensitivity. However, whether such products are safe for subjects with other types of gluten hypersensitivity such as life-threatening systemic anaphylaxis remains to be established.

### 5.4. Optimized Thermal Processing Methods May Be Used to Produce Potentially Hypoallergenic Gluten Products

Four exciting studies have reported the effects of thermal processing on gluten allergenicity using in vitro methods. One such study by Lupi et al., 2019, examined the impact of boiling (100 °C) on the allergenic properties of purified alcohol-soluble glutens [78]. The study employed a wheat flour extract of total gliadins, further isolating α-gliadins through a reversed-phase high-performance liquid chromatography method. IgE-based dot blotting was performed using pooled serum from a cohort of wheat-allergic subjects (composed of five groups containing specific IgE antibodies against gliadin ranging from 27 ng/mL to 167 ng/mL in the testing). Mast cell degranulation effects of the gliadin were tested in vitro via cell-line assay. The findings indicated a complete loss of IgE reactivity and mast-cell degranulation potential in boiled gliadins. However, the validation of non-allergenicity in animal models and humans remains a crucial step.

In contrast to the above findings, another study by Pastorello et al., 2007, reported that boiling wheat flour had no significant impact on IgE reactivity [8]. They obtained serum samples from 22 wheat-allergic subjects and tested each sample individually using the IgE Western blot method. The discrepancy between the two papers suggests that whereas purified gliadins are susceptible to boiling, gliadins contained in wheat flour appear to retain their IgE reactivity despite boiling. 

A study conducted by De Angelis et al., 2007, compared the impact of pepsin and pancreatin digestion on the allergenicity of two types of breads to simulate the effect of gastric and intestinal digestion on allergenicity of wheat proteins: (i) conventional yeast bread and (ii) an experimental sourdough bread created using a cocktail of selected lactic acid bacteria plus yeast named as VSL#3 [79]. The study focused on allergenicity testing of albumins, globulins, and gliadins extracted from the enzyme-digested breads. The results revealed that enzyme treatment of VSL#3 sourdough bread demonstrated markedly diminished IgE reactivity of albumins, globulins, and gliadins compared to conventional bread. This suggests the intriguing possibility that allergens contained in VSL#3 sourdough bread may be easily degraded by the guy enzymes and therefore better tolerated by wheat allergic individuals compared to the conventional yeast bread. The study’s noteworthy strength lies in presenting a novel method for producing potentially hypo-/nonallergenic sourdough bread.

Finally, Kobayashi et al., 2005, delved into testing the impact of soy sauce production steps on the allergenicity of both non-gluten and gluten allergens [63]. Notably, the initial stage of soy sauce production involves the high-temperature roasting of wheat. Leveraging pooled serum from five children allergic to wheat as a source of anti-wheat IgE antibodies, they conducted ELISA testing. The results revealed a significant (32%) reduction in gluten allergens following the roasting and cracking of wheat (Figure 2). 

In summary, these findings suggest the following effects of thermal processing on gluten allergenicity: (i) baking temperatures used in bread making have the potential to reduce gluten allergenicity; (ii) boiling has different effects on gluten allergenicity depending on whether gluten proteins are in pure form or contained within the wheat flour matrix; and (iii) roasting and cracking steps during soy sauce production have the potential to reduce gluten allergenicity. Future testing using preclinical rodent models and human clinical testing are needed to confirm these effects. 

## 6. Conclusions and Future Directions

Extensive animal pre-clinical testing and a limited human pre-clinical study have identified at least five promising dietary-based novel therapeutic possibilities for gluten hypersensitivity: (i) fermentation; (ii) probiotics; (iii) enzyme hydrolysis; (iv) thioredoxin, phosphorylation, and deamidation; and (v) gene targeting of gluten loci in wheat plants. However, there are only two human clinical testing of such novel therapeutics reported in the literature.

Our findings collectively suggest the following research agenda to advance translational science in gluten hypersensitivity:

(i) Challenges and opportunities to consider in animal model testing: preclinical dose–response studies in animal models using quantitative readouts of disease phenotypes of gluten hypersensitivity noted in humans are urgently needed; although few studies have investigated models of anaphylaxis, more work is needed to carefully test the effects on validated quantifiable readouts of life-threatening systemic anaphylaxis, such as hypothermic shock responses, mucosal mast cell degranulation responses, and histamine responses. Furthermore, mechanisms of effects of treatments on gluten hypersensitivity also need to be investigated. All animal models testing the effects of novel therapeutics have utilized adjuvants to create gluten hypersensitivity in their studies. Recent research shows that mechanisms underlying adjuvant-based versus adjuvant-free models of food allergy can be substantially different [80]; since human gluten allergenicity is pathogenesis typically does not involve adjuvants such as alum, adjuvant-free models might be more helpful in developing therapeutics for human application. Adjuvant-free models of gluten hypersensitivity reported recently can be employed for this purpose [60,61]; furthermore, most animal models have focused on systemic anaphylaxis only; animal models of other types of hypersensitivities caused by gluten (atopic dermatitis, baker’s asthma, etc.) are urgently needed for preclinical testing of novel diet-based therapeutics in development (Figure 1).

(ii) Challenges and opportunities to consider in humans: most studies have focused on testing binding of IgE antibodies to altered gluten proteins. Since IgE binding only demonstrates sensitization at best and not the elicitation of clinical reactions, more work is needed to develop methods for clinical testing in humans or humanized cell lines/humanized animal models; only two in vivo studies of novel therapeutics (use of wheat naturally deficient in ω5 gliadin, such as Einkorn for WDEIA; and enzyme-treated wheat flour for gluten-induced AD) have been reported for human gluten hypersensitivity [69,70]. Furthermore, most human preclinical and clinical research has been focused on WDEIA and AD as the types of gluten hypersensitivity disorders for developing diet-based therapeutics. Therefore, more translational research is needed for other promising approaches in gluten hypersensitivity disorders not only for AD and WDEIA but also for all other allergic conditions including life-threatening systemic anaphylaxis, typical gluten-induced food allergy (vomiting, diarrhea, and urticaria), airways/conjunctival allergies, and baker’s asthma caused by gluten (Figure 1).

In summary, evidence from the current literature demonstrates at least five promising dietary-based therapeutic approaches for mitigating gluten hypersensitivity that are in various stages of research and development. Further progress in future translational research promises potential novel dietary-based therapeutic options for the management of gluten hypersensitivity in humans.

## Figures and Tables

**Figure 1 ijms-25-04399-f001:**
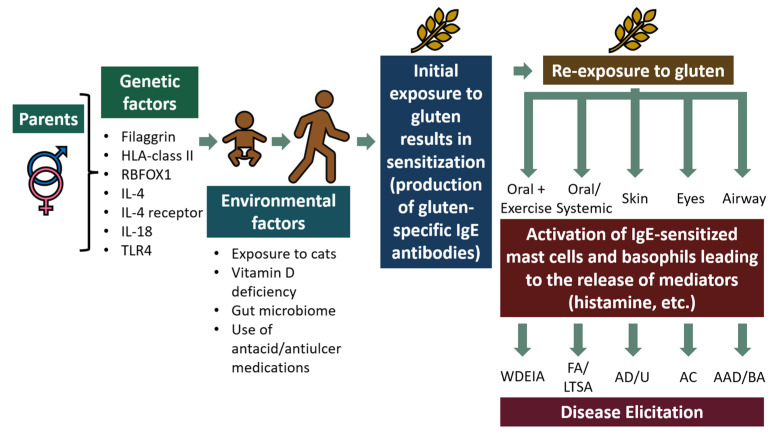
Pathogenesis of gluten hypersensitivity: the role of genetics and environmental factors. Inheritance of susceptibility gene variants from parents renders the offspring a propensity to develop atopic sensitization to gluten that is modulated by co-exposure to environmental factors. Some of the known genetic and environmental factors are illustrated in the figure. Re-exposure to gluten results in the development of gluten hypersensitivity reactions. The routes of sensitization can be oral, skin, eyes, and airways. The routes of disease elicitation can be oral, skin, eyes, airways, and blood transfusion. Exercise upon ingestion of gluten can result in WDEIA within one to four hours. WDEIA: wheat-dependent exercise-induced anaphylaxis; FA: food allergy; LTSA: life-threatening systemic anaphylaxis; AD: atopic dermatitis; U: urticaria; AC: allergic conjunctivitis; AAD: allergic airways disease; BA: baker’s asthma; IgE: immunoglobulin E.

**Figure 2 ijms-25-04399-f002:**
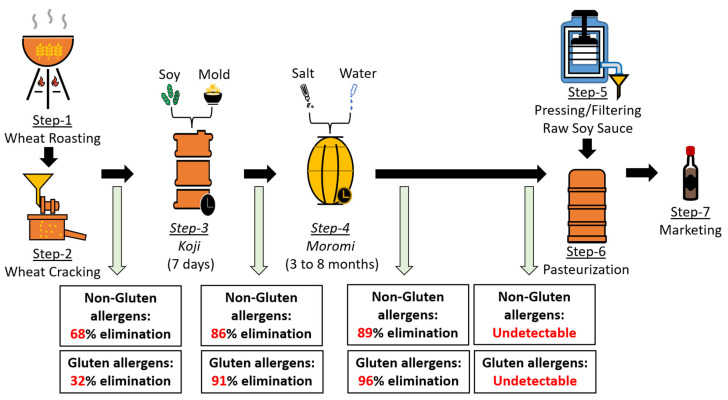
Key steps in traditional soy sauce production and the progressive reduction in and elimination of gluten and non-gluten allergens. This figure shows the various steps involved in soy sauce production. The effects on gluten and non-gluten allergens based on the research reported in the literature are also summarized. The salt-soluble non-gluten allergen content as measured by direct ELISA is progressively reduced by approximately 68%, 86%, 89%, and undetectable by the end of step 2, *koji*, *moromi* day 10, and *moromi* day 48, respectively, as measured by direct ELISA. The salt-insoluble wheat allergen (gluten) content as measured by direct ELISA is progressively reduced by approximately 32%, 91%, 96%, and undetectable by the end of step 2, *koji*, *moromi* day 10, and *moromi* day 48, respectively, as measured by direct ELISA.

**Table 1 ijms-25-04399-t001:** Potential of thioredoxin and heat-killed *Listeria monocytogenes* for treating gluten hypersensitivity in a dog model.

Model and Therapy	Protein Used	Sensitization Phenotype	Disease Phenotype
Mitigation of gluten (wheat) allergenicity using thioredoxin treated wheat flour [45]	Wheat gluten	Specific IgE	Wheal and flair reaction upon SPT
Mitigation of gluten (wheat) allergenicity using HKL treatment of dogs [46]	Wheat flour	Specific IgE	Increase in minimum dose require to elicit positive SPT reaction

Abbreviations used in the table: HKL: heat-killed listeria; SPT: skin prick test; IgE: Immunoglobulin E.

**Table 2 ijms-25-04399-t002:** Potential of gluten gene editing, deamidation, and enzyme treatment for gluten hypersensitivity in rat and guinea pig models.

Model and Therapy	Protein Used	Sensitization Phenotype	Disease Phenotype
Injection of 1BS-18 gluten lacking ω-5 gliadin reduced anaphylaxis in BN rats [47]	Gluten from Hokushin and 1BS-18 wheat	Specific IgE	HSR decreased
Induction of oral tolerance by early ingestion of 1BS-18 gluten that is lacking ω-5 gliadin in BN rats [48]	Commercial gluten, gluten prepared from Hokushin, and 1BS-18 wheat flours	Specific IgE	HSR decreased
Induction of oral tolerance using deamidated gliadin in BN rats [49]	Gliadin and deamidated gliadin	Specific IgE	None
Enzyme-treated gluten reduced airway allergenicity in BN rats [50]	Gluten and hydrolyzed gluten with cellulase and actinase	Specific IgE	Reduced BAL immune cells (neutrophils, lymphocytes, and eosinophils)
Mitigation of oral allergy using 1BS-18 gluten lacking ω-5 gliadin in Guinea Pigs [51]	Commercial gluten and 1BS-18 gluten	None	Significant decrease in allergy scores

Abbreviations used in the table: BN: Brown Norway Rat; HSR: hypothermic shock response; BAL: broncho-alveolar lavage; IgE: Immunoglobulin E; 1BS-18: deletion in the 1B short arm.

**Table 3 ijms-25-04399-t003:** Potential of probiotics, enzymatic hydrolysis, deamidation, and phosphorylation in mouse models of gluten hypersensitivity.

Model and Therapy	Protein Used	Sensitization Phenotype	Disease Phenotype
Mitigation of gluten anaphylaxis using oral administration of *Pediococcus acidilactici* XZ31 in Balb/c mice [52]	Commercial wheat gluten	Specific IgE	Clinical symptom scores of systemic anaphylaxis
Enzyme-hydrolyzed gluten reduced sensitization in Balb/c mice [53]	Hydrolyzed wheat gluten with AN-PEP	Specific IgE	None
Oral treatment with L-arabinose reduced anaphylaxis symptoms in Balb/c mice [54]	Commercial gliadin	Total IgE	Clinical symptom scores of systemic anaphylaxis, allergic enteritis (histology, jejunum)
Hydrolysis and deamidation of gliadin reduced sensitization in Balb/c mice [55]	Hydrolyzed and deamidated gliadin	Specific IgE	None
Phosphorylated gliadin, and enzyme treatment of gliadin reduced sensitization in Balb/c mice [56]	Gliadins, phosphorylated gliadins, hydrolyzed gliadins with alcalase or papain	Specific and total IgE	None
Repeated oral administration of deamidated gliadin reduced allergenicity in native gliadin sensitized Balb/c mice [57]	Native gliadin and deamidated gliadin by carboxylation cation exchange resin	Specific IgE	Blood elevation of histamine

Abbreviations used in the table: AN-PEP: *Aspergillus niger*-derived prolyl endopeptidase; IgE: Immunoglobulin E.

**Table 4 ijms-25-04399-t004:** Potential of fermentation, gluten gene targeting, deamidation, thioredoxin, and enzyme treatment for gluten hypersensitivity in humans.

Model and Therapy	Protein Used	Sensitization Phenotype	Disease Phenotype
In vitro model, soy sauce fermentation reduced/eliminated gluten allergenicity [63]	Non-gluten and gluten	Specific IgE	None
In vitro model, gene targeting to remove conventional gluten reduced gluten allergenicity [64,65,66,67]	Gluten from gene targeted wheats using various techniques (Gene Translocation, Gene inactivation, gene deletion and gene silencing)	Specific IgE	None
In vitro model, thioredoxin treatment of gluten reduced its allergenicity [68]	Alcohol-soluble gluten extract was treated with thioredoxin	Specific IgE	None
In vivo and in vitro model, diploid genotype (AA) reduced gluten allergenicity [69]	Alcohol (40%) soluble gluten extract	Specific IgE	Negative SPT reaction in WDEIA 13/14 patients
In vivo and in vitro model, enzyme hydrolyzed wheat flour reduced gluten allergenicity [70]	Enzyme hydrolyzed wheat flour	Specific IgE	13/15 AD patients tolerated cupcakes made from treated flour: 2/15 developed severe urticaria

Abbreviations used in the table: SPT: skin prick test; AD: atopic dermatitis; WDEIA: wheat-dependent exercise-induced anaphylaxis; IgE: Immunoglobulin E.

**Table 5 ijms-25-04399-t005:** Diversity of microorganisms in different types of soy sauce based on country of origin.

Soy Sauce Country of Origin	Mold *	Yeast *	Gram-Positive Bacteria *	Gram-Negative Bacteria *
**Korea** [71]	*Tetrapisispora* *Cryptococcus* *Penicillium* ***Aspergillus* sp., *A. flavus:* (6/6)**	*Wickerhamomyces*	Not Studied	Not Studied
		*Torulaspora*		
		*Tetrapisispora*		
		*Rhodotorula*		
		*Pichia*		
		*Microbotryum*		
		*Debaryomyces*		
		** *Candida:* ** ** (6/6)**		
		*Zygosaccharomyces*		
**China** [72]	** *Aspergillus* **	*Starmerella* *Wickerhamiella* *Saturnispora* ** *Candida* **	*Weisella* ***Bacillus:* (3/5)** ***Lactobacillus:* (3/5)** *Leuconostoc* *Lactococcus* *Pediococcus* *Enterococcus* *Micrococcus* *Streptococcus* ***Staphylococcus:* (3/5)** *Propionibacteriacea*	*Xanthomonas* *Salmonella* *Pseudomonas* *Pantoea* *Lebsiella* *Dechloromonas* *Cupriavidus* *Arsenophonus* *Acidobacteriaceae*
**Japan** [73]	** *Aspergillus: A. oryzae* ** ** *Geotrichum* **	*Zygosacchormyces* ** *Candida etchellsii* ** ** *C. nodaensis* ** ** *C. versatilis* ** ** *C. catenulata* ** *Wickerhamomyces* *Pichia* *Trichosporon*	*Weisella* ** *Lactobacillus* ** ** *Staphylococcus gallinarum* ** ** *S. xylosus* ** ** *S.arlettae* ** ** *S. saprophyticus* ** ** *S. succinus* ** ** *S. cohnii* ** ** *S. caprae* ** ** *S. kloosii* ** *Pediococcus* *Tetragenococcus*	Not studied
**Not specified** [74]	** *Aspergillus: Aspergillus sojae* ** ** *A. parasiticus* ** *Peronospora*	*Sacchoramycopsis* *Millerozyma* *Pichia* ** *Candida* ** ** sp** **.** ** *C. rugosa* ** ** *C. orthopsilosis* ** ** *C. tropicalis* **	** *Staphylococcus* ** *Kurthia* ** *Bacillus* ** *Paenibacillus* *Corynebacterium*	*Klebsiella* ** *Enterobacter* **
**China (LSSF)** [75]	** *Aspergillus oryzae* **	*Wickerhmomyces* *Saccharomycopsis* *Kluyveromyces* ** *Candida rugosa* ** ** *C. glabrata* ** ** *C. tropicalis* ** *Pichia* *Trichosporon*	*Weisella* ** *Bacillus subtilis* ** ** *B. licheniformis* ** ** *B. pumilus* ** ** *Staphylococcus sciuri* ** ** *S. gallinarum* ** ** *S. succinus* ** ** *S. aureus* ** ** *S. cohnii* ** *Corynebacterium* *Kurthia* *Enterococcus* ** *Lactobacillus* ** *Rothia* *Arhrobacter* *Pediococcus*	*Escherichia* ** *Enterobacter* **
**China** **(Xianshi)** [76]	** *Aspergillus niger* ** *Cladosporium* *Fusarium* *Lichtheimia* *Absidia*	*Meyerozyma* ***Candida*** *parapsilosis* *Sterigmatomyces*	***Bacillus amyloliquefaciens*** ***B. subtilis*** ***B.**lincheniformis*** ***B. methylotrophicus*** ***B. aerius*** ***B. halmapalus*** ***B. flexus*** ***B. thuringiensis*** ***B. coagulan*** *Scopulibacillus* *Shimwellia* *Weissella* *Lactococcus* *Clostridium* *Streptomyces* *Microlunatus*	*Klebsiella* *Pantoea* ** *Enterobacter* ** *Erwinia* *Trichodesmium*

* Bold-faced mold, yeast, and bacteria are the organisms used most in soy sauce preparation independent of country of origin.

## Data Availability

All data are provided in the paper.

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
