# Peer review of "Advances in Gluten Hypersensitivity: Novel Dietary-Based Therapeutics in Research and Development"

_ijms, 2024, doi:10.3390/ijms25084399_

Round 1

Reviewer 1 Report

Comments and Suggestions for Authors

 This is a review collecting the current information to produce hypoallergenic wheat products. The two articles which have been recently published should be included in this analyses and discussed: Tsurunaga Y, Arima S, Kumagai S, Morita E. Low allergenicity in processed wheat products using Tannins from agri-food wastes. Foods 202Jul 17;12(14):2722. doi: 10.3390/foods12142722, and Zitong YuUral YunusbaevAllan FritzMichael TilleyAlina AkhunovaHarold TrickEduard AkhunovCRISPR-based editing of the ω- and γ-gliadin gene clusters reduces wheat immunoreactivity without affecting grain protein quality. Plant Biotechnol J 2023 Nov 17.pp1-12, DOI: 10.1111/pbi.14231.

Minor points to be corrected

Point 1: Page 5 line 5; Buchanon should be correcred as Buchanan.

Point 2: Page 6 line 2; use of use of should be use of.

Author Response

Jorgensen et al., 2024 (Manuscript ID – 2819542)

IJMS Reviewer comments

Reviewer 1 comments

This is a review collecting the current information to produce hypoallergenic wheat products.

The two articles which have been recently published should be included in this analyses and discussed:

Tsurunaga Y, Arima S, Kumagai S, Morita E. Low allergenicity in processed wheat products using Tannins from agri-food wastes. Foods 2023 Jul 17;12(14):2722. doi: 10.3390/foods12142722,

and Zitong Yu, Ural Yunusbaev, Allan Fritz, Michael Tilley, Alina Akhunova, Harold Trick, Eduard Akhunov. CRISPR-based editing of the ω- and γ-gliadin gene clusters reduces wheat immunoreactivity without affecting grain protein quality. Plant Biotechnol J 2023 Nov 17.pp1-12, DOI: 10.1111/pbi.14231.

Response: Excellent suggestion! We have included the two articles mentioned and have written the following in the section: “4. Other approaches to produce hypoallergenic gluten products”

Tsurunaga et al., 2023 recently released an exciting report [59]. They sought to investigate the potential reduction in wheat allergenicity through the addition of tannins from chest-nut inner skin (CIS) or young persimmon fruit (YPF) to wheat flour. Cookies made with cake flour were prepared, with tannins accounting for 3%, 5%, and 10% of the total ingredient weight. Both CIS and YPF were chosen for their high tannin content. The evaluation of wheat allergen content involved the use of two ELISA kits, which demonstrated a notable reduction in protein content, especially gliadin and other wheat proteins, with CIS treatment showing more marked effects than YPF treatment. Immunoblotting with a polyclonal rabbit anti-ω5-gliadin IgG antibody further supported these findings, revealing a decrease in IgG immunoreactivity in both types of treatments. Despite the strength of demonstrating reduced immunoreactivity, this study has certain limitations. IgE-binding was not investigated and considering the IgE-mediated nature of wheat allergy, further research is essential in this regard to determine the effects on allergenicity. Additionally, the immune reactivity study focused solely on ω-5 gliadin, necessitating future exploration of the tannin treatment effects on other wheat allergens to comprehensively understand its potential impact. Furthermore, future in vivo testing using preclinical animal models will be necessary as a crucial step preceding eventual human clinical trials [60,61].

Yu et al., 2023 investigated whether it was feasible of modifying specific segments of gluten-encoding genes to substantially diminish wheat immunotoxicity without adversely affecting the physiochemical properties crucial for breadmaking [62]. The hypothesis, though implicit, postulated that the multiplexed CRISPR-Cas9 editing of the common wheat (Triticum aestivum) Fielder cultivar (originally released by the University of Idaho in 1974), aimed at modifying ω- and γ-gliadin genes, would result in reduced immunoreactivity compared to the non-edited counterpart. To implement this approach, seven gRNAs were designed to edit the gliadin gene, with three targeting ω-gliadins on chromosomes 1A and 1D and four targeting ω-gliadins on chromosome 1B. Monoclonal antibodies R5 and G12, known for their high predictability of wheat immunotoxicity in gluten-sensitive patients, were employed to assess immunotoxicity through ELISAs. The findings indicated a significant reduction in immunoreactivity in the edited wheat line compared to the non-edited line. Despite the strength of demonstrating reduced immunoreactivity, the need for further investigations remains, particularly examining IgE-reactivity (allergenicity) in these edited lines. Further in vivo testing in preclinical animal models would allow for the evaluation of CRISPR-Cas9 editing’s effect on potentially reduced clinical allergic reactivity. This step is essential before progressing to clinical trials involving human subjects [60,61].

Minor points to be corrected

Point 1: Page 5 line 5; Buchanon should be corrected as Buchanan.

Response: Great catch. We have corrected the manuscript as per your suggestion.

Point 2: Page 6 line 2; use of use of should be use of.

Response: Thank you for pointing this out. We have corrected this mistake in the manuscript.

We would like to thank you for your time and efforts in providing these excellent comments to improve the scientific merit of our paper.

Reviewer 2 Report

Comments and Suggestions for Authors

The authors have extensively reviewed the literature about new strategies based on diet as a possible therapy for gluten hypersensitive patients. The topic is interesting and I believe is a notable contribution for the field. The manuscript is well organized and it is easy to read, tables and figures are useful and necessary.

Just a comment about the title in section 4: in that section is explained basically in vitro evaluations of gluten hyersensitivity, it is true that the articles cited used normally human sera, but I consider that authors should be more cautious with the title (Human testing). 

Authors explain in detail strategies to obtain hypoallergenic wheat ingredients, specifically enzymatic  treatments, but they not mention (or from my point of view, not enough) other thermal processing that has been studied to reduce the capacity to ellicitate IgE release in vitro. Why the authors decided not to explain more about these strategies? It is true that they made an effort to illustrate the soy production process, that includes some thermal processing, but I think they should include other preliminary studies in this context. 

Author Response

Jorgensen et al., 2024 (Manuscript ID – 2819542)

IJMS Reviewer comments

Reviewer 2 comments

The authors have extensively reviewed the literature about new strategies based on diet as a possible therapy for gluten hypersensitive patients. The topic is interesting and I believe is a notable contribution for the field. The manuscript is well organized and it is easy to read, tables and figures are useful and necessary.

Response: Thank you for your interest in this work. We appreciate your support for this manuscript.

Just a comment about the title in section 4: in that section is explained basically in vitro evaluations of gluten hypersensitivity, it is true that the articles cited used normally human sera, but I consider that authors should be more cautious with the title (Human testing).

Response: Great point! We have worded the section as follows: Effects of fermentation, gluten gene targeting, deamidation, thioredoxin, and enzyme treatment on gluten allergenicity in humans.

Authors explain in detail strategies to obtain hypoallergenic wheat ingredients, specifically enzymatic treatments, but they not mention (or from my point of view, not enough) other thermal processing that has been studied to reduce the capacity to ellicitate IgE release in vitro. Why the authors decided not to explain more about these strategies? It is true that they made an effort to illustrate the soy production process, that includes some thermal processing, but I think they should include other preliminary studies in this context.

Response: Great suggestion! The following section has been added to the manuscript:  

5.4 Optimized thermal processing methods may be used to produce potentially hypoallergenic gluten products

There are four exciting studies reporting the effects of thermal processing on gluten allergenicity using in vitro methods. One such study by Lupi et al., 2019 examined the impact of boiling (100°C) on the allergenic properties of purified alcohol-soluble glutens [78]. The study employed a wheat flour extract of total gliadins, further isolating α-gliadins through a reversed phase high-performance liquid chromatography method. IgE-based dot blotting was performed using pooled serum from a cohort of wheat-allergic subjects (composed of five groups containing specific IgE antibodies against gliadin ranging from 27 ng/mL to 167 ng/mL in the testing). Mast cell degranulation effects of the gliadin were tested in vitro via cell line assay. The findings indicated a complete loss of IgE reactivity and mast cell degranulation potential in boiled gliadins. However, the validation of non-allergenicity in animal models and humans remains a crucial step.  

In contrast to the above findings, another study by Pastorello et al., 2007 reported that boiling wheat flour had no significant impact on IgE reactivity [8]. They obtained serum samples from 22 wheat-allergic subjects and tested each sample individually using the IgE western blot method. The discrepancy between the two papers suggests that whereas purified gliadins are susceptible to boiling, gliadins contained in wheat flour appear to retain their IgE reactivity despite boiling.

A study conducted by De Angelis et al., 2007, compared impact of pepsin and pancreatin digestion on the allergenicity of two types of breads to simulate the effect of gastric and intestinal digestion on allergenicity of wheat proteins: i) conventional yeast bread; and ii) an experimental sourdough bread created using a cocktail of selected lactic acid bacteria plus yeast named as VSL#3 [79]. The study focused on allergenicity testing of albumins, globulins, and gliadins extracted from the enzyme digested breads. The results revealed that enzyme treatment of VSL#3 sourdough bread demonstrated markedly diminished IgE reactivity of albumins, globulins, and gliadins compared to conventional bread. This suggests the intriguing possibility that allergens contained in VSL#3 sourdough bread may be easily degraded by the guy enzymes, and therefore better tolerated by wheat allergic individuals compared to the conventional yeast bread. The study's noteworthy strength lies in presenting a novel method for producing potentially hypo-/nonallergenic sourdough bread.

Finally, Kobayashi et al., 2005, delved into testing the impact of soy sauce production steps on the allergenicity of both non-gluten and gluten allergens [61]. Notably, the initial stage of soy sauce production involves the high-temperature roasting of wheat. Leveraging pooled serum from five children allergic to wheat as a source of anti-wheat IgE antibodies, they conducted ELISA testing. The results revealed a significant (32%) reduction in gluten allergens following the roasting and cracking of wheat (Figure 2).

In summary, these findings suggest the following effects of thermal processing on gluten allergenicity: i) baking temperatures used in bread making have the potential to reduce gluten allergenicity; ii) boiling has different effects on gluten allergenicity depending on whether gluten proteins are in pure form or contained within the wheat flour matrix; and iii) roasting and cracking steps during soy sauce production have the potential to reduce gluten allergenicity. Future testing using preclinical rodent models and human clinical testing are needed to confirm these effects. 

We thank you for your efforts and time for providing this excellent feedback to improve our paper.

Reviewer 3 Report

Comments and Suggestions for Authors

This is a very good manuscript about gluten hypersensitivity with a clearly stated aim and conclusion.

However, it would be beneficial if the introduction section described how authors searched the databases, what keywords were used, how many articles were found, etc.…

Table 2: Not all the abbreviations are explained (1BS-18 or IgE). In general, each figure is separate and should include a description, and all the words should be explained.

The fond on figures could be bigger. It should be almost the same size as text.

Table 5 is a bit hard to read because there are no lines that discriminate between bacteria in different studies.

Author Response

Jorgensen et al., 2024 (Manuscript ID – 2819542)

IJMS Reviewer comments

Reviewer 3

This is a very good manuscript about gluten hypersensitivity with a clearly stated aim and conclusion.

Response: Thank you for these kind comments. We appreciate your support of this manuscript.

However, it would be beneficial if the introduction section described how authors searched the databases, what keywords were used, how many articles were found, etc.…

Response: This is a great suggestion. We have added the following has been added to the introduction section:

We employed various combinations of the following keywords for our search using the PubMed and Google Scholar databases: gluten, hypersensitivity, therapy, in vivo, in vitro, IgE, wheat, hypoallergenicity, animal model, dog, rat, guinea pig, mice, human, and dietary-based. The PubMed search retrieved articles ranging from 2 to 1209; the Google Scholar search retrieved hits ranging from 919 to 1920. Only relevant English language articles were retrieved and used to address the above objectives. All articles chosen for the study are included in the references. The focus of this study was specifically on IgE-mediated gluten hypersensitivity, necessitating the exclusion of articles addressing non-IgE mediated gluten disorders (including celiac disease, non-celiac gluten sensitivity, and eosinophilic disorders).

Table 2: Not all the abbreviations are explained (1BS-18 or IgE). In general, each figure is separate and should include a description, and all the words should be explained.

Response: Excellent suggestions. We have gone through the manuscript and ensured that each abbreviation has been explained.

The fond on figures could be bigger. It should be almost the same size as text.

Response: Very good suggestion. We have increased the font on all of the figures. It is now in line with the size of the text.

Table 5 is a bit hard to read because there are no lines that discriminate between bacteria in different studies.

Response: Great point, we have added lines within this table to make it easier to discriminate between the various studies.

We would like to thank you for your outstanding feedback to improve our manuscript. We appreciate your time and efforts.

Round 2

Reviewer 1 Report

Comments and Suggestions for Authors

The revised manuscript was adequately corrected according to my comments.